

# Description and classification of bivalve mollusks hemocytes: a computational approach

Yuriy A. Karetin[1,2], Aleksandra A. Kalitnik[1,2], Alina E. Safonova[2] and Eduardas Cicinskas[2]

[1] National Scientific Center of Marine Biology of Far Eastern Branch of the Russian Academy of Sciences, Vladivostok, Russia
[2] Far Eastern Federal University, Vladivostok, Russia

## ABSTRACT

The fractal formalism in combination with linear image analysis enables statistically significant description and classification of "irregular" (in terms of Euclidean geometry) shapes, such as, outlines of *in vitro* flattened cells. We developed an optimal model for classifying bivalve *Spisula sachalinensis* and *Callista brevisiphonata* immune cells, based on evaluating their linear and non-linear morphological features: size characteristics (area, perimeter), various parameters of cell bounding circle, convex hull, cell symmetry, roundness, and a number of fractal dimensions and lacunarities evaluating the spatial complexity of cells. Proposed classification model is based on Ward's clustering method, loaded with highest multimodality index factors. This classification scheme groups cells into three morphological types, which can be distinguished both visually and by several linear and quasi-fractal parameters.

## INTRODUCTION

The quantitative characterization and classification of fibroblast-like cells with complex irregular shapes is a challenge. Finding a solution to this will facilitate identification of cell types at different differentiation stages and help investigate their morphogenetic transformations, both *in vivo* and *in vitro*. Unlike cells with clearly recognizable and typified structural elements (e.g., neurons (*Pushchin & Karetin, 2009*)), invertebrate immune cells (bivalve hemocytes and echinoderm coelomocytes) do not have unambiguously recognizable structural elements of the external morphology. There are many transitional forms among filopodia, lamellipodia, and pseudopodia, and the boundary between the cell body and the processes is not always clear. This complicates description and classification of these cells using conventional structural parameters.

Parameters of quasi-fractal organization are used to describe "irregular" and chaotic (in terms of the Euclidean geometry) biological patterns with the *de facto* standard (*Dokukin et al., 2015*). This standard treats a natural pattern as a quasi-fractal that can be analyzed with a number of fractal dimensions and spatial heterogeneity and lacunarity. In this work, we applied a complex approach combining classical morphometric parameters with

Corresponding author
Eduardas Cicinskas,
cicinskas@gmail.com

quasi-fractal ones for comparative classification of hemocytes and coelomocytes of bivalve mollusks and echinoderms, which enabled identification of morphotypes of adhered immune cells, which are characteristic of each animal species.

## MATERIAL AND METHODS

The study was performed on 458 hemocytes of the bivalve mollusk *Spisula sachalinensis* (*Bivalvia*, *Mactridae*) (Schrenck, 1862) and 628 hemocytes of the bivalve mollusk *Callista brevisiphonata* (*Bivalvia*, *Veneridae*) (Carpenter, 1865). The animals were collected in the Vostok Bay (Peter the Great Bay, Sea of Japan). Cells were collected as previously described (*Karetin, 2016*; *Karetin & Pushchin, 2017*). Briefly, hemolymph was collected from the cardiac sac onto coverglass and incubated at room temperature for 1 h. Afterwards cells were fixed in a 4% formalin solution and stained with hematoxylin-eosin. Photographs of flattened cells were taken with Zeiss Axiovert 200M Apotome microscope, then they were sketched by hand and converted into a one-bit format for further analysis.

8 quasi-fractal parameters (prefactor lacunarity (**LCFD PreLac**), prefactor lacunarity heterogeneity or translational invariance (**outLCFD PreLac**), mean mass fractal dimension (**MMFD**), mean mass dimension of images contour (**outMMFD**); mean local connected fractal dimension of contour images of cells (**outMeanLCFD**); mean local fractal images contour dimension (**outMeanLFD**); mean dimensions of images contour (**outMeanD**), lacunarity L (**LF**)) and 9 linear parameters (cell perimeter (**Per**), circularity (**Circ**), hull's circularity (**Hull'sCirc**), Roundness (**Round**), hull's center of mass radii Min/Max ratio (**M/MHull'sCM**), ratio of Min/Max distance to circle's center (**M/MRadCirc**)), aspect ratio (**AR**), Max half division (**1/2half**) and inner/outer bounding circle ratio (**in50/out50**) out of 39 linear and non-linear parameters were chosen for final analysis. They were calculated using FracLac 2.5 plug-in for ImageJ 1.41 and Photoshop SC3. Full lists of parameters their explanation and computation can be viewed at Supplemental Information 1. All parameters were normalized by subtracting the mean and dividing the result by the standard deviation to equalize their contribution to the classification as cluster variables.

STATISTICA 12.0 and NCSS 2007 software packages were used for statistical analysis. Correlation among parameters was measured using the Pearson's linear correlation analysis. Highly correlated parameters, presumably describing close morphological properties, were excluded from the analysis. When choosing from two highly correlated parameters, the preference was given to a parameter with a higher multimodality index showing whether the distribution of parameter values was mono-, bi-, or multimodal (*Schweitzer & Renehan, 1997*). Cell were classified using hierarchical cluster analysis with the Ward's clustering algorithm and the Euclidean distance as a measure of dissimilarity. The dimension of multiparametric space was reduced using the factor analysis. A variance analysis was used to verify the cluster structure.

## RESULTS

Table 1 provides a description of all the measured variables. We used Pearson's linear correlation analysis to exclude highly correlated parameters from analysis. 31–34 cell

**Table 1  Measured linear and quasi-fractal parameters.**

| *Spisula sachalinensis* | | | *Callista brevisiphonata* | | |
|---|---|---|---|---|---|
| **Variable** | **Mean** | **SE** | **Variable** | **Mean** | **SE** |
| Area | 258175,9 | ±111077,8 | Area | 240782,8 | ±5314,675 |
| Per | 4823,5 | ±2496,7 | Per | 5792,4 | ±117,301 |
| Circ. | 0,139 | ±0,0839 | Circ | 0,090 | ±0,0028 |
| Hull'sCirc | 0,829 | ±0,0759 | Hull's Circ | 0,812 | ±0,0030 |
| Round | 0,636 | ±0,1551 | Round | 0,646 | ±0,0066 |
| M/MHull'sCM | 1,674 | ±0,3627 | M/M Hull'sCM | 1,697 | ±0,0151 |
| M/MRadCirc | 1,608 | ±0,5666 | M/M Rad Circ | 1,713 | ±0,0241 |
| 1/2 half | 2,312 | ±2,7958 | 1/2 half | 7,068 | ±0,3938 |
| in 50/out 50 | 0,951 | ±0,7683 | in 50/out 50 | 1,811 | ±0,0514 |
| AR | 1,692 | ±0,5326 | AR | 1,686 | ±0,0231 |
| LCFD PreLac | 0,456 | ±0,3212 | LCFD PreLac | 7,502 | ±0,2034 |
| outLCFD PreLac | 0,059 | ±0,0176 | outLCFD PreLac | 0,043 | ±0,0006 |
| MMFD | −1,984 | ±0,0043 | MMFD | −1,975 | ±0,0004 |
| outMMFD | −1,038 | ±0,0111 | out MMFD | −1,027 | ±0,0003 |
| outMeanLCFD | 1,033 | ±0,0098 | out Mean LCFD | 1,023 | ±0,0003 |
| outMeanLFD | 1,080 | ±0,0237 | out Mean LFD | 1,102 | ±0,0011 |
| outMeanD | 1,177 | ±0,0617 | out Mean D | 1,186 | ±0,0018 |
| LF | 0,815 | ±0,2648 | LF | 1,167 | ±0,0133 |

**Table 2  Weakly correlated parameters ranked by the multimodality index.**

| *Spisula sachalinensis* | | *Callista brevisiphonata* | |
|---|---|---|---|
| **Selected parameters** | **Multimodality index** | **Selected parameters** | **Multimodality index** |
| LCFD PreLac | 0,705279674 | AR | 0,649614849 |
| 1/2 half | 0,690093836 | M/Mrad Circ | 0,621260648 |
| Per | 0,649817151 | Circ | 0,612524578 |
| M/MradCirc | 0,582812559 | in 50/out 50 | 0,576105304 |
| in 50/out 50 | 0,571990685 | Per | 0,575294522 |
| AR | 0,533511041 | 1/2 half | 0,503101471 |
| M/Mhull'sCM | 0,496496281 | MMFD | 0,463013064 |
| Hull'sCirc | 0,460160702 | out Mean LCFD | 0,40903434 |
| MMFD | 0,379435841 | LCFD PreLac | 0,370674562 |
| outLCFD PreLac | 0,370878375 | – | – |
| outMeanLFD | 0,342725457 | – | – |

parameters for each species had a significantly high ($p < 0.05000$) correlation with one or more other parameters. For further analysis of the cell morphology of both species, we selected weakly correlated ($r < 0.7$) or uncorrelated parameters with the highest multimodality index (Table 2).

One of the main cluster analysis problems is the so-called "curse of dimension", which means that the quality of clustering decays rapidly as the model dimensionality (the number of parameters) increases (*Gordon, 1999*; *Xu & Wunsch, 2008*).

To further reduce the number of parameters, we used two approaches: selection of uncorrelated parameters with a multimodality index above a given threshold and the factor analysis; in this case, both factors and parameters loading the factors were used for classification. Among parameters loading each factor, parameters with the maximum multimodality index were also selected (*Pushchin & Karetin, 2014*).

Factors of the factor analysis were chosen using the Varimax method of orthogonal rotation of the main factor axes. The Varimax method maximizes the spread of load squares for each factor, which increases large values of factor loads and decreases small values of factor loads.

In *C. brevisiphonata* and *S. sachalinensis*, we identified 5 and 4 factors, respectively, which were significantly loaded with at least one parameter (Tables 3 and 4). The first factor in both species was loaded with fractal dimensions and lacunarities of different types and, in general, reflected quasi-fractal characteristics of cell morphology. The second factor was loaded with parameters reflecting size characteristics of the cell, such as the area, perimeter, and sizes of the cell bounding circle and convex hull. The third factor was loaded with parameters describing the roundness and elongation of the cell and its convex hull (Round, AR, Hull'sCirc). The fourth factor in both species was loaded with parameters associated with the local fractal dimension and mass dimension of contour cell images: (outMeanLCFD, outMeanLFD, outLCFD PreLac, and outMMFD). Therefore, loads of most parameters in both species were similarly distributed over four factors determining the main characteristics of cell morphology. In addition, the fifth factor in *C. brevisiphonata* included quasi-fractal parameters of contour images; in *S. sachalinensis*, these parameters were combined with quasi-fractal parameters of silhouette images in the first factor. However, Explained Variation eigenvalues of the fourth factor in *C. brevisiphonata* dropped below 3; apart from this factor, Explained Variation values in both species decreased below 3 only in factors lacking any parameter significantly loading the factors. To formally and uniformly limit the number of used parameters, only factors with an Explained Variation value above 3 and significantly loaded with at least one parameter were used in the cluster analysis. In each species, these requirements were met by four factors that were used as 4 parameters for clustering. In addition, as parameters chosen for the cluster analysis, we used parameters with the maximum multimodality index, which loaded each factor (*C. brevisiphonata*: AR, Per, MMFD, outMeanD; *S. sachalinensis*: AR, Per, LF, outLCFD PreLac).

We chose the Ward's clustering algorithm and the Euclidean distance as an intercluster difference. This clustering technique provided the best results in classification of neurons (*Pushchin & Karetin, 2009*) and invertebrate immune cells (*Karetin & Pushchin, 2015*). The number of tested clusters did not exceed the number of cell types normally present in the immune system of invertebrates (*Dyrynda, Pipe & Ratcliffe, 1997*; *Chang, Tseng & Chou, 2005*; *Ladhar-Chaabouni & Hamza-Chaffai, 2016*). Also, we tested cluster sets with the intercluster communication distance that visually significantly exceeded the distances between subsequent bifurcations of cluster divisions.

Differences between clusters were estimated using discriminant analysis techniques, including the Mahalanobis intercluster distance estimation (*Mahalanobis, 1936*) and *F*-test

**Table 3 Factor analysis of the morphological parameters of *Callista brevisiphonata*.** Marked loadings are >.70,0000.

| Variable | Factor loadings (Varimax raw) extraction: principal axis factoring | | | | |
|---|---|---|---|---|---|
| | Factor 1 | Factor 2 | Factor 3 | Factor 4 | Factor 5 |
| Area | 0,898344 | 0,081324 | 0,017790 | 0,040363 | 0,145123 |
| Circ | −0,245306 | −0,011567 | −0,853883 | −0,173950 | 0,103754 |
| AR | −0,024839 | −0,177094 | 0,027887 | 0,053657 | -0,691950 |
| Hull'sCirc | 0,248542 | 0,523053 | −0,060343 | −0,124967 | 0,558290 |
| M/MHull'sCM | −0,097370 | −0,889826 | −0,078575 | −0,012798 | −0,266480 |
| M/M RadCirc | −0,095711 | −0,765280 | 0,007187 | 0,024369 | −0,220561 |
| outMeanD | 0,437328 | 0,117268 | 0,734259 | −0,000113 | 0,018558 |
| Var in Count | 0,828276 | 0,221451 | 0,183255 | 0,200946 | 0,153856 |
| LCFD PreLac | 0,323227 | −0,027924 | 0,284534 | 0,086889 | −0,022068 |
| MeanMassFD | −0,431402 | −0,072897 | 0,634385 | 0,096541 | −0,192108 |
| outMeanLCFD | −0,122948 | 0,016813 | −0,092966 | −0,985520 | 0,043433 |
| outMeanLFD | 0,163686 | 0,015080 | 0,627006 | 0,054636 | 0,173220 |
| outMeanMassFD | 0,111875 | −0,008582 | 0,064887 | 0,987433 | −0,051194 |
| 1/2half | 0,718833 | −0,043749 | 0,159371 | 0,200869 | −0,148511 |
| in50/out50 | −0,390528 | −0,099130 | 0,233359 | 0,089379 | 0,170615 |
| Area out | 0,760254 | 0,054657 | 0,513314 | 0,188803 | −0,046116 |
| Round out | 0,059278 | 0,386290 | −0,067658 | −0,078121 | 0,850117 |
| Expl.Var | 3,421268 | 1,922556 | 2,550114 | 2,147736 | 1,813877 |
| Prp.Totl | 0,201251 | 0,113092 | 0,150007 | 0,126337 | 0,106699 |

for equality of variances. In our case, the Mahalanobis distance defines the distance between obtained clusters in a multidimensional space of variables. The $F$-test, or the Fisher's test, is used to compare the variances of two normally distributed populations. When more than two populations are compared, the $F$-test is calculated as the inter-sample variance of the all cluster centroids is compared with the combined in all clusters intra-sampling variance.

The use of factors as parameters gave a spread in the mean Mahalanobis distances for solutions with a different number of cell clusters of both species, ranging between 2 and 7, as many cell types are usually selected in immune cells of invertebrate species close to the studied ones; the same approach yielded F values lying between 113 and 140. Testing of 4 parameters with the highest multimodality value, without using factor analysis, gave a spread in the mean Mahalanobis distances of 9–19, with the Fisher statistics values occurring between 155 and 196. For cells of both species, the use of parameters loading factors gave the Mahalanobis distances between 11 and 17 and F values between 177 and 218. According to the results of F statistics, the best solution for both species was a 3-cluster structure constructed on the basis of 4 parameters loading factors: 218.17 for cells of *C. brevisiphonata* and 216.98 for cells of *S. sachalinensis* (Fig. 1, Table 5). The Mahalanobis distances for these cluster solutions also were among the highest ones (Table 6).

Based on the Lambda quotient values, the contribution of all four parameters to the classification of *S. sachalinensis* and *C. brevisiphonata* cells was high, without dominance or
**Table 4  Factor analysis of the morphological parameters of *Spisula sachalinensis*.** Marked loadings are >.70,0000.

| Variable | Factor loadings (Varimax raw) extraction: principal axis factoring | | | | |
|---|---|---|---|---|---|
| | Factor 1 | Factor 2 | Factor 3 | Factor 4 | Factor 5 |
| Area | −0,40982 | 0,769811 | 0,274500 | 0,078471 | 0,102884 |
| Circ. | −0,83180 | −0,263856 | 0,053532 | −0,222871 | −0,047215 |
| AR | 0,03529 | 0,024757 | −0,824474 | −0,050441 | −0,037324 |
| Round out | −0,06194 | −0,002724 | 0,904602 | 0,043889 | 0,014734 |
| Var in Count | −0,09494 | 0,812251 | 0,386541 | 0,040463 | −0,167519 |
| Hull'sCirc | −0,18412 | 0,138136 | 0,763370 | 0,152132 | 0,191975 |
| M/MHull'sCM | −0,07944 | −0,189913 | −0,453385 | −0,130423 | 0,071628 |
| M/M RadCirc | 0,20104 | −0,156081 | −0,429464 | −0,094532 | 0,010053 |
| outMeanD | 0,77702 | 0,258795 | 0,211161 | 0,248320 | 0,329420 |
| LCFD PreLac | 0,04799 | 0,048295 | −0,008780 | 0,294671 | 0,179491 |
| MMFD | −0,13560 | −0,672211 | 0,097037 | 0,178344 | 0,608486 |
| outMeanLCFD | −0,16340 | −0,250238 | 0,079762 | 0,883067 | 0,109141 |
| outMeanLFD | 0,58175 | −0,014001 | 0,089297 | 0,393662 | 0,170759 |
| outMMFD | 0,14229 | 0,247899 | −0,077718 | −0,900644 | −0,101673 |
| Per | 0,50941 | 0,716218 | 0,219273 | 0,020457 | 0,274903 |
| in50/out50 | −0,25912 | 0,057885 | 0,090105 | 0,030264 | −0,111232 |
| 1/2 half | −0,00923 | 0,101447 | −0,132909 | −0,000644 | −0,091831 |
| Expl.Var | 10,54201 | 8,113110 | 3,938550 | 3,558619 | 2,149974 |
| Prp.Totl | 0,28492 | 0,219273 | 0,106447 | 0,096179 | 0,058107 |

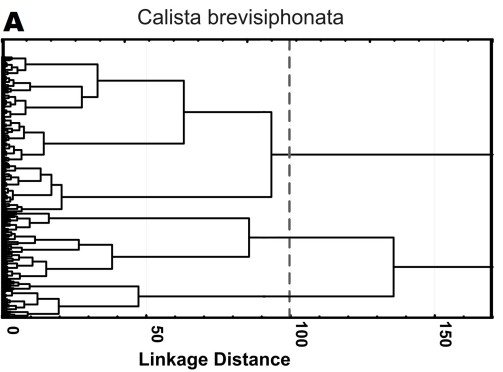
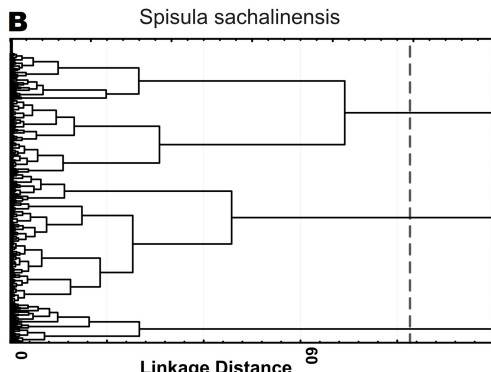

**Figure 1  Hierarchical cluster analysis of hemocytes from *C. brevisiphonata* and *S. sachalinensis*, which is based on parameters with the highest multimodality index, loading each of four factors of the factor analysis.** Clustering algorithm: the Ward's method, a measure of the intercluster difference: the Euclidean distance, a 3-cluster solution.

**Table 5** *F*-test values for the difference in the cluster variance of the optimal cluster solution (**Fig. 1**).

| | *C. brevisiphonata; F*-values; $df = 4,622$ | | |
| --- | --- | --- | --- |
| | G_1:1 | G_2:2 | G_3:3 |
| G_1:1 | | $p = 0,00$ | $p = 0,00$ |
| G_2:2 | 219.9391 | | $p = 0,00$ |
| G_3:3 | 232.1484 | 202.4672 | |
| | Average *F*-test value for all clusters 218.17 | | |
| | *S. sachalinensis; F*-values; $df = 4,451$ | | |
| | G_1:1 | G_2:2 | G_3:3 |
| G_1:1 | | $p = 0,00$ | $p = 0,00$ |
| G_2:2 | 232.5024 | | $p = 0,00$ |
| G_3:3 | 277.5630 | 140.8834 | |
| | Average *F*-test value for all clusters 216,98 | | |

**Table 6** Squared Mahalanobis distances between clusters of the optimal cluster solution (**Fig. 1**).

| | *C. brevisiphonata* | | |
| --- | --- | --- | --- |
| | G_1:1 | G_2:2 | G_3:3 |
| G_1:1 | 0.00000 | 15.13759 | 12.74839 |
| G_2:2 | 15.13759 | 0.00000 | 7.16835 |
| G_3:3 | 12.74839 | 7.16835 | 0.00000 |
| | Average Mahalanobis distances between clusters 11,7 | | |
| | *S. sachalinensis* | | |
| | G_1:1 | G_2:2 | G_3:3 |
| G_1:1 | 0.00000 | 19.66674 | 23.90871 |
| G_2:2 | 19.66674 | 0.00000 | 5.75293 |
| G_3:3 | 23.90871 | 5.75293 | 0.00000 |
| | Average Mahalanobis distances between clusters 16,43 | | |

exclusion of any of the parameters (Table 7); in *C. brevisiphonata*, the AR parameter had a somewhat more significant contribution (Table 7).

## DISCUSSION

Three morphological types of hemocytes from *C. brevisiphonata* identified using 3 clusters of the selected cluster solution differ in several parameters of the linear and nonlinear morphology. Type 1 includes cells which are characterized by values of the lacunarity (LF) and mean fractal dimension of contour images (outMeanD) significantly higher than those of types 2 and 3 describing cells with a complex structure of boundaries, most unevenly filling the space. Cells of this type are visually characterized by an elongated shape with an average number of long processes. Type 2 is represented by cells with the highest outLCFD PreLac value as well as medium, but significantly different from other types in outMeanD and perimeter values (Fig. 2). Cells of this type are characterized by a less elongated shape and a larger number of smaller processes. The third type includes cells with the simplest microsculpture of boundaries characterized by the minimum outMeanD value and the

**Table 7** The Tukey-Kramer multiple-comparison test for the difference in mean values of selected parameters of three morphological hemocyte types from *S. sachalinensis* and *C. brevisiphonata*, which are identified based on the optimal cluster solution.

| | | *C. brevisiphonata* | | |
|---|---|---|---|---|
| Parameters | Wilks's lambda | Partial-Lambda | *F*-remove-(3,621) | *p*-value |
| MMFD | 0.163939 | 0.692703 | 91.8292 | 0.000000 |
| Per | 0.142717 | 0.795709 | 53.1454 | 0.000000 |
| outMeanD | 0.145625 | 0.779820 | 58.4460 | 0.000000 |
| AR | 0.280243 | 0.405223 | 303.8301 | 0.000000 |
| | | *S. sachalinensis* | | |
| Parameters | Wilks's lambda | Partial-Lambda | *F*-remove-(2,451) | *p*-value |
| LF | 0.177949 | 0.746466 | 76.5902 | 0.000000 |
| Per | 0.174506 | 0.761195 | 70.7449 | 0.000000 |
| AR | 0.171701 | 0.773631 | 65.9827 | 0.000000 |
| outLCFD PreLac | 0.234937 | 0.565398 | 173.3339 | 0.000000 |

largest AR value which are characteristic of cells with an almost round shape. Cells of this type are characterized by the most "regular" symmetrical shape with a low number of relatively small processes (Fig. 3).

The cluster model for classification of hemocytes from *S. sachalinensis*, using parameters of four factors, also includes three cell types (Fig. 4). The first type includes cells with the highest outMeanD, LF, and perimeter values among the identified types. This characterizes cells of complex morphology with a high number of large and small processes. Cells of the second type, like the second type cells from *C. brevisiphonata*, have the highest LCFD PreLac value for contour images, and an average, but significantly different outMeanD value. These cells have a visually simpler "average" shape with a smaller number of small processes. Type 3 includes cells with a smaller, compared to other types, area of spreading (Area), the highest AR value, and a low outMeanD value (Fig. 5). Visually, these cells have the simplest microsculpture of boundaries with the minimum number of processes, but the general shape of these cells is diverse and asymmetric. Usually, cells of this type have an elongated, sometimes slightly round shape, which is typical of moving cells.

According to the Tukey-Kramer multiple-comparison test for the mean value difference, the identified cell types significantly differed from each other both in most classifying parameters (Table 8) and in several other parameters of the linear and quasi-fractal morphology.

Therefore, although the best classification for cells of each of the studied species includes different parameters and distinguishes species-specific types differing in various aspects of morphology, the optimal classification structure of immune cells of both species uses a common algorithm including a Ward's hierarchical classification and using parameters with the highest multimodality index, loading factors in the factor analysis.

A classification model with the best cluster structure combines both linear and quasi-fractal parameters, thereby reflecting various aspects of the cell morphology. Due to loading of various factors of the factor analysis with parameters describing similar morphological

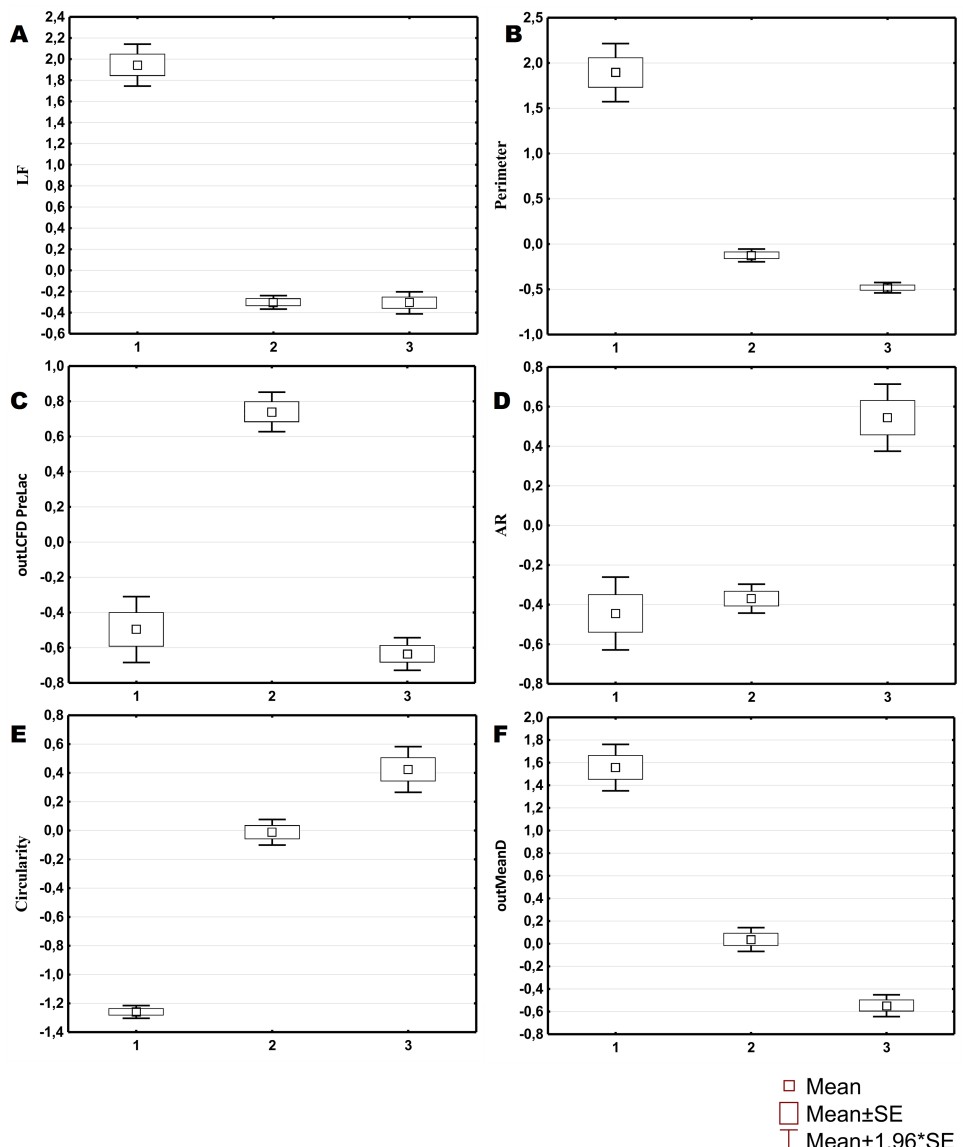

**Figure 2** Average values and SE of the main parameters used for the description of cell types from *C. brevisiphonata.*

aspects, the classification of cells of the studied species comprises simultaneously parameters reflecting the cell asymmetry (AR), size characteristics (Per), and quasi-fractal parameters describing the structural complexity of silhouette (MMFD) and contour (outMeanD) cell images. In this case, among parameters representing each morphological aspect, we chose a parameter with the maximally multimodal distribution over a sample, which indicates heterogeneity of the sample in this parameter and most clearly reveals the sample composite structure that includes cells of different morphological types.

In biological terms, the main issue of the morphology of cells from various animal species is what determines the difference in the morphology of flattened hemocytes from

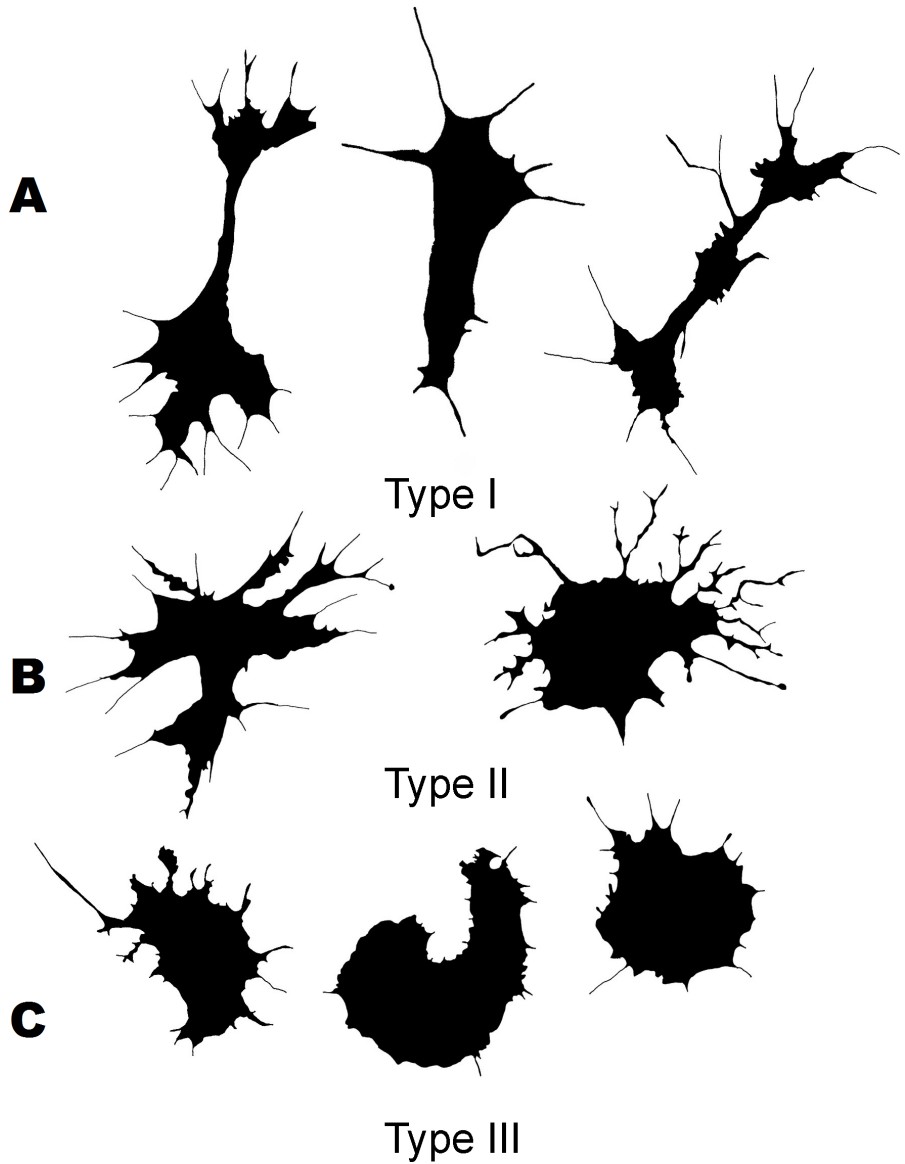

A

Type I

B

Type II

C

Type III

**Figure 3** Silhouette images of cells from *C. brevisiphonata*.

different invertebrate species? The initial hypothesis suggests searching for an ecological aspect of functional differences manifesting in the morphology of cells and leading to the difference in cell shapes among separate species. However, we found no significant ecological differences between the studied species. Both mollusk species live in overlapping areas, and both species are filter feeders digging into the sea floor.

Regardless of whether environmental causes underlie the differences in shapes of hemocytes and coelomocytes from different species, and whether the specific features of their morphology are affected by natural selection, it is obvious that the difference in cell shapes is genetically associated with certain differences in cell physiology: the

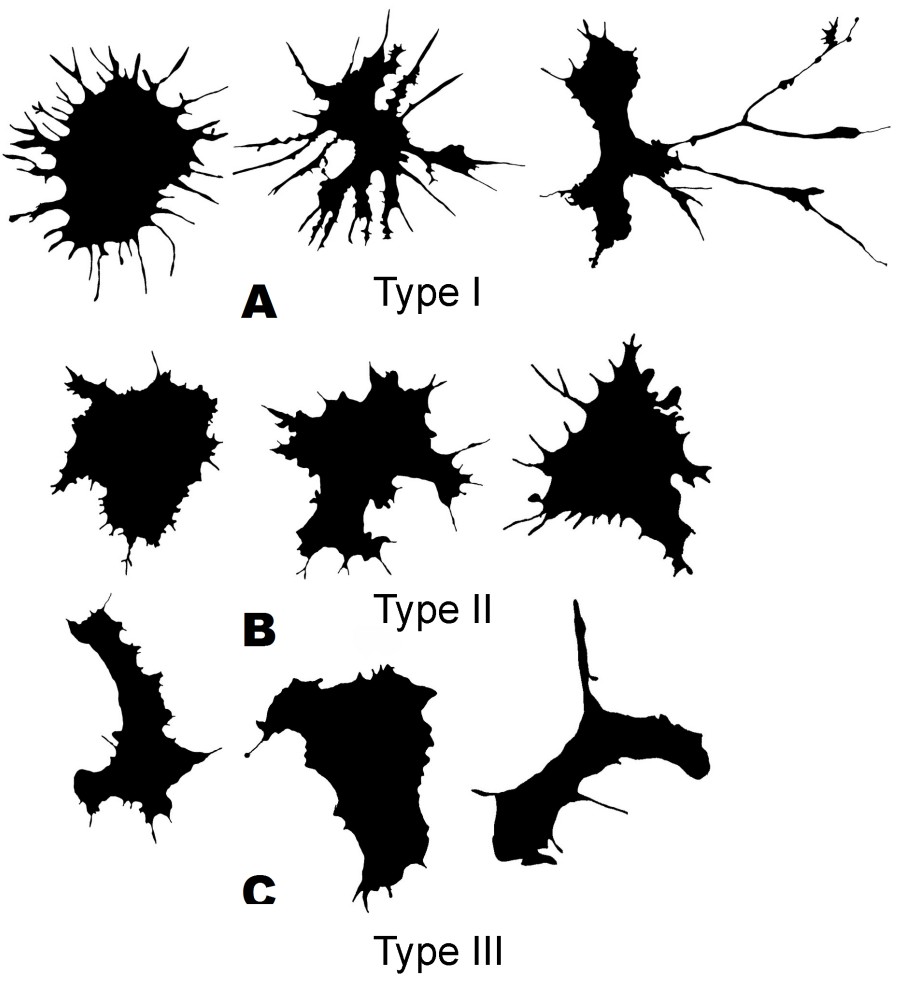

**Figure 4** Silhouette images of cells from *S. sachalinensis*.

general cytoskeleton structure, cell behavior (*Pomp et al., 2018*), etc. Therefore, it is logical to consider cell shapes in terms of biological analysis of parameters that describe cell shapes, which would answer the question: what kind of genetic and cytophysiological species-specific features of the cell determine the value of a certain parameter. This will make it possible to predict features of cell physiology based on detailed morphological analysis.

## CONCLUSIONS

The optimal classification that is based on the morphological features of immune cells of the studied bivalve species uses a common universal algorithm that includes a Ward's hierarchical cluster analysis based on parameters with the highest multimodality index, loading factors of the factor analysis. This cluster structure demonstrates the highest $F$ statistics values for intercluster variance differences.

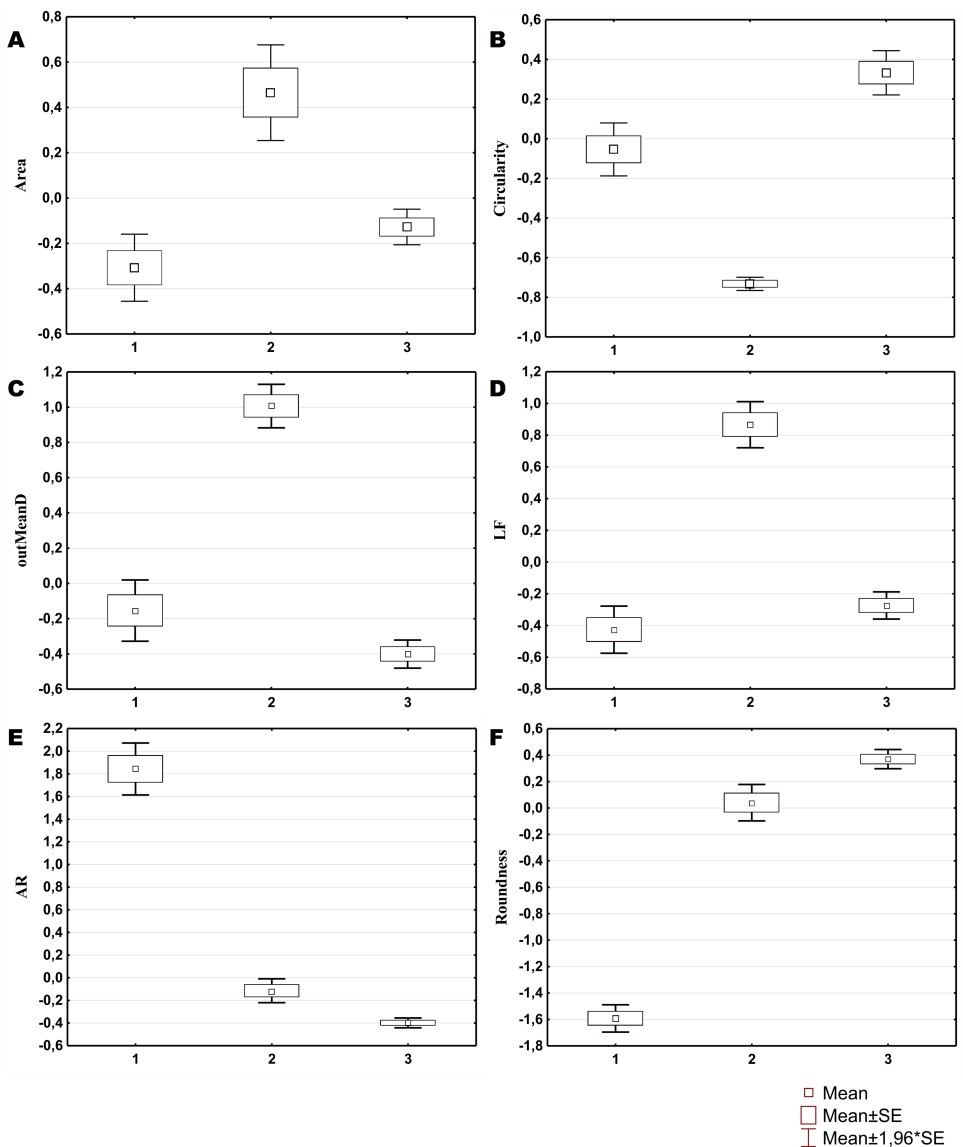

**Figure 5** Average values and SE of the main parameters used for the description of cell types from *S. sachalinensis.*

The identity of the optimal classification algorithm for cells of both species with very close values of discriminant functions describing interspecies differences in the variances of cell morphotypes in the selected classification model suggests a common morphological structure of hemocytes, despite interspecific differences, both in numerical values of used parameters and in the set of parameters selected for the classification. Therefore, the cell classification algorithm may be recommended as an optimal method for morphological classification of hemocytes adhered to a two-dimensional substrate.

**Table 8  Wilks' Lambda for the optimal cluster solution.**

| S. sachalinensis | | | C. brevisiphonata | | |
|---|---|---|---|---|---|
| Parameters | Cell type | Inter-type difference | Parameters | Cell type | Inter-type difference |
| LF | 3 | 1 | MMFD | 3 | 1, 2 |
| | 2 | 1 | | 1 | 3, 2 |
| | 1 | 3, 2 | | 2 | 3, 1 |
| Per | 3 | 2, 1 | Per | 3 | 2 |
| | 2 | 3, 1 | | 1 | 2 |
| | 1 | 3, 2 | | 2 | 3, 1 |
| AR | 1 | 3 | AR | 3 | 2, 1 |
| | 2 | 3 | | 2 | 3, 1 |
| | 3 | 1, 2 | | 1 | 3, 2 |
| outLCFD PreLac | 3 | 2 | Area | 1 | 2 |
| | 1 | 2 | | 3 | 2 |
| | 2 | 3, 1 | | 2 | 1, 3 |
| | 3 | 2, 1 | | 3 | 1, 2 |
| Area | 2 | 3 | outMeanD | 1 | 3, 2 |
| | 1 | 3 | | 2 | 3, 1 |

### Funding
This work was supported by Russian Foundation for Basic Research (RFBR) (No. 18-04-00430A). The funders had no role in study design, data collection and analysis, decision to publish, or preparation of the manuscript.

### Grant Disclosures
The following grant information was disclosed by the authors:
Russian Foundation for Basic Research (RFBR): 18-04-00430A.

### Competing Interests
The authors declare there are no competing interests.

### Author Contributions
- Yuriy A. Karetin conceived and designed the experiments, performed the experiments, analyzed the data, contributed reagents/materials/analysis tools, prepared figures and/or tables, authored or reviewed drafts of the paper, approved the final draft.
- Aleksandra A. Kalitnik performed the experiments, contributed reagents/materials/analysis tools, authored or reviewed drafts of the paper.
- Alina E. Safonova performed the experiments.
- Eduardas Cicinskas performed the experiments, analyzed the data, contributed reagents/materials/analysis tools, prepared figures and/or tables, authored or reviewed drafts of the paper, approved the final draft.

## Data Availability

The raw data is available at figshare: https://figshare.com/projects/Description_and_ classification_of_bivalve_mollusks_hemocytes/64136.

## Supplemental Information

Supplemental information for this article can be found online at http://dx.doi.org/10.7717/ peerj.7056#supplemental-information.

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
