# Peer review of "Description and classification of bivalve mollusks hemocytes: a computational approach"

_PeerJ, doi:10.7717/peerj.7056_

## Round 0.1 · original submission · Major Revisions

Both the reviewers suggested major revision. However I recommend resubmit revised manuscript to PeerJ taking into account all the comments..

Reviewer 1 ·

Basic reporting

Subject is important. Taking into account geometrical complexity of objects, methods of fractal geometry are appropriate.

Experimental design

Experimental methods are adequate.

Validity of the findings

Three morphological cell types, obtained by cluster analysis methods, from a biological point of view look very reasonable.

Additional comments

However the statistical analysis is quite weak.

The results of the factor analysis are not presented either in the tables or in the graphs; a verbal description is not enough.

An attempt to statistically justify the results of cluster analysis contains a vicious circle – it is based on the same data as cluster analysis. It’s mistakenly to compute significance by such a way. If we take a random data set and apply the cluster analysis, it will always break the set of objects into some groups. If we apply discriminant analysis namely to these groups using the same traits, DA will always show significant differences.

A possible way out of this situation would be bootstrap analysis, but in general for cluster analysis these methods have not yet been developed. Therefore, it is necessary to confirm the results obtained by means of biological and geometrical arguments.

Concrete remarks:

Annotation, 18, 106. “dimensional characteristics (area, perimeter)”.
In physics and mathematics, the dimension of a mathematical space (or object) is informally defined as the minimum number of coordinates needed to specify any point within it (Wikipedia). It isn’t size. Fractal dimension is a fractional number.

66 – 68. All ratios are not linear, but dimensionless parameters.

71. “were normalized …” specify, please. There are several variants of normalization with equal contributions to clustering.
80 «… Euclidean distance as a measure of proximity». Euclidean distance isn’t a measure of proximity, but is a measure of dissimilarity.

116 “… dropped below 3”. What is the measure of explained variation? Eigenvalues or percents?

135 – 137 “When more than two populations are compared, the F-test is calculated as the external variance to internal variance ratio, determining the difference between variances of the selected clusters.”
This is not true. No variances of clusters are compared. But the inter-sample variance of the all cluster centroids is compared with the combined in all clusters intra-sampling variance.

P-values aren't provided in tables 3, 4. The tables are symmetric, p-values can be placed below or above the diagonal.

The cell types shown in Fig. 2 aren't reflected in Fig. 1.

Conclusion. Article should be remade.

Reviewer 2 ·

Basic reporting

The manuscript entitled “Description and classification of bivalve mollusks hemocytes:a computational approach” submitted by the authors Y. A. Karetin, A. A. Kalitnik, A. E. Safonova and E. Cicinskas aims to describe the hemocyte populations from bivalves, Spisula sachalinensis and Callista brevisiphonata by a new method for identification.
The manuscript is generally well-written and organized; however there are several general concerns, which, in my opinion, do not allow publishing it.

Experimental design

1. In Introduction the authors did not provide available information about types of hemocytes of the species studied. If there are no such data for these bivalves it should be stated.
2. The analysis is carried out on 458 hemocytes of Spisula sachalinensis and 628 hemocytes of Callista brevisiphonata. Is this number of cells sufficient for statistics and further morphometric measurements? According to the Table 1 the SE for some cells characteristics is more than ½ of the mean value (see the values of Area, Per, 1/2half, in50/out50, LCFD PreLac for Spisula sachalinensis and 1/2half for Callista brevisiphonata). Moreover, some of these parameters were used for the further characterization of hemocytes. This means, that determination of cell types, which are hardly distinguished due to their similarity and occurrence of transitional forms, is based on the parameters with great variability. May be authors should provide data confirming the significance of the differences in the characteristics chosen between types of hemocytes for each species.

Validity of the findings

In Discussion the authors do not provide any comparative analysis of the results obtained by the new approach and other methods used for hemocyte identification. Thus, it remains unclear, whether the new technique gives the adequate precise distinguishing of hemocytes. There are no citations of literature within the section, therefore, the reader does not have an opportunity to independently come to conclusion about the efficacy of the method. The authors should provide information, which agrees with the data, obtained by the new method and confirms its convenience of application.
References. The authors should substantially work with the literature. Among 12 citations only a half can be classified as “new” or “latest” works. The rest of references were published before 2008, more than 10 years ago. Most of the “new” citations belong to the author of the manuscript. Thus, for my opinion, the results of the research are insufficiently justified.

Additional comments

These are the reasons why to reject this publication as it stands. I recommend to resubmit a new MS with extra data linked to the evidence of the efficacy of the new approach.

---

## Round 0.2 · accepted · Accept

The manuscript has been significantly improved. As I see, the reviewers' suggestions were take into account. The paper would be accepted in current form.

Technical note - title of Table 8 is sideway of the table - please check it at final submission.